# Spatial Distribution, Source Analysis and Health Risk Study of Heavy Metals in the Liujiang River Basin in Different Seasons

**DOI:** 10.3390/ijerph192315435

**Published:** 2022-11-22

**Authors:** Shi Yu, Wanjun Zhang, Xiongyi Miao, Yu Wang, Rongjie Fu

**Affiliations:** 1Key Laboratory of Karst Dynamics, MNR & GZAR, Institute of Karst Geology, Chinese Academy of Geological Sciences, Guilin 541004, China; 2International Research Center on Karst under the Auspices of UNESCO, Guilin 541004, China; 3College of Geography and Environmental Science, Northwest Normal University, Lanzhou 730070, China

**Keywords:** heavy metals, anion, cation, anthropogenic activity, river water, risk assessment

## Abstract

Three high-frequency sampling and monitoring experiments were performed at the Lutang and Luowei transects of the Liujiang River entrance and at the southeast exit of the Liuzhou during 2019 for the purpose of assessing physico-chemical variables and human health hazards of water heavy metals in different rainfall processes. There were significant seasonal variations in concentrations of 11 heavy metals and most variables showed higher levels during the dry season. The distribution of heavy metals in the Liuzhou area varied significantly by region. Pollution source analysis indicated distinct seasons of wetness and dryness. The dry season is dominated by anthropogenic activities, while the wet season is dominated by natural processes. The results of hazard quotient (HQ) and carcinogenic risk (CR) analysis showed that the health risk of non-carcinogenic heavy metals in the wet season is slightly higher than that in the dry season. Seasonal changes in carcinogenic risk are the opposite; this is due to the combined influence of natural and human activities on the concentration of heavy metals in the river. Among them, Al was the most important pollutant causing non-carcinogenic, with As being a significant contributor to carcinogenic health risk. Spatially, the downstream Luowei transect has a high health risk in both the dry and rainy seasons, probably due to the fact that the Luowei transect is located within a major industrial area in the study area. There are some input points for industrial effluent discharge in the area. Therefore, high-frequency monitoring is essential to analyze and reduce the heavy metal concentrations in the Liujiang River during dry and wet seasons in order to protect the health of the residents in the area.

## 1. Introduction

With the development of the economy and society, natural (bed rock and mineral deposits) and man-made processes release more heavy metals into aquatic ecosystems [1,2,3,4,5]. Therefore, due to their toxicity, persistence, and bioaccumulation, heavy metals in water environments such as surface water and drinking water sources that exceed certain concentrations and pose a risk to human health are of great concern worldwide [6,7,8]. Many studies have reported increased health risks from excessive exposure to Cd, Cr, As, Pb, and Hg through water ingestion. In addition, As is perhaps the most concerning metalloid in drinking water because of its high toxicity and widespread occurrence in the environment [9,10,11]. Drinking water poses a great danger to safety, especially in terms of human health.

The sources of heavy metal and water chemical pollution in rivers are related to regional geology, precipitation runoff, contaminated sediment relase, atmospheric deposition, and human activities [12,13,14,15,16,17]. Currently, methods such as correlation analysis, redundancy analysis (RDA), and principal component analysis (PCA) are widely used to analyze the sources of heavy metals in surface water [18,19,20]. Each river has its own characteristics due to differences in geography, climate, and the degree of urbanization. As a relatively unique ecological environment system in the geographic environment [21], karst rivers exhibit strong sensitivity to changes in the natural environment or the effects of human activities, and many studies have reported that the weathering of carbonate rocks may be related to metal contamination in karst areas of southwest China [22,23,24]. Consequently, the combination of anions and cations produced by rock weathering and dissolution with heavy metals allows a better assessment of the sources of heavy metal pollution in river water.

China is one of the countries with the most extensive karst distribution area. One quarter of the world’s karst area is distributed in China. If calculated according to the distribution area of carbonatite, it can reach 7.463 million km^2^, accounting for about 1/3 of country’s land area. The limestone distribution in Guangdong and Yunnan is concentrated and extensive, accounting for about half of the country’s distribution area. Guangxi is famous for its unique karst landforms. The Liujiang River, located in Liuzhou City, Guangxi Province, is representative of the subtropical karst basin in Southwest China. The extensive industrial operation in Liuzhou has made heavy metals the main pollutants in the Liujiang River [25,26]. In previous studies, heavy metals (except mercury) were found to be below the standard of three types of surface water quality [26]. The presence of Cr and As in the Liujiang River highlighted potential health risks [26,27]. Therefore, it is necessary to conduct a health risk assessment of heavy metals in the Liujiang River to assess the related health risks to humans. Ions and heavy metals in water are mainly affected by rock weathering and dissolution, rainfall, agriculture, and domestic sewage discharge [28]. Liuzhou city is located in the north–central part of Guangxi and has a mid-subtropical monsoon climate, with high temperature and rain in summer and less in winter. Differences in rainfall between seasons cause certain dynamic changes in the water chemical content and heavy metals in water. Drastic precipitation in the wet season may aggravate the fluctuation of water chemistry and heavy metals in the Liujiang River and then significantly elevate the heavy metals in water and impact water environment safety. The aim of this study was to (1) assess the linkage changes in heavy metals, anions, and cations in river water in each season; (2) analyze the possible sources of heavy metals in wet and dry seasons; and (3) determine the risk to human health resulting from potential human exposure to heavy metals in water from both transects. This work is expected to provide scientific evidence for water environment risk management in the Liuzhou area.

## 2. Materials and Methods

### 2.1. Study Area

The Liujiang River Basin is located at 108°32′ and 110°28′ east between 23°54′ and 26°03′ north in northern and southwestern Guangxi (Figure 1). The total length is 272 km. It starts at the intersection of the Rongjiang River and the Longjiang River in Fengshan town, flows through most of Liuzhou city, then restarts from the Lutang section in the northwest of the city (upstream, subscript “R1”), enters the urban area, passes through the urban area of Liuzhou and flows out along the Luowe section (downstream, subscript “R2”), and then wraps the northern peninsula of Liuzhou city into a pot shape (thus explaining why Liuzhou city is also known as “hot city”). Liujiang is the largest river in Liuzhou. Xunjiang, Duliujiang, Longjiang, and Rongjiang are all located in the upper reaches of Liujiang. The mining industry is widely distributed, resulting in generally high levels of heavy metals in river water [12]. There are certain industrial parks and residential areas in the district. Therefore, setting sampling points at the starting and ending points from Liujiang to Liuzhou can comprehensively reflect the possible impacts of natural and human activities on the water quality of Liuzhou in the urban area of Liuzhou and the upper reaches of the river. The annual average temperature is 18 °C~20 °C and the annual precipitation is 1400~11800 mm. Depending on the hydrology of the basin, the wet season runs from April to September. Eighty percent of the annual precipitation falls during the wet season, which subsequently dilutes river pollutants, especially in industrial areas, and continuous rainfall also causes heavy metals from river bottom sediments to enter river water bodies. Liuzhou is an important industrial town in Guangxi, and it is also a key city for the emission of “three wastes” in Guangxi. Ninety-two percent of the industrial, agricultural, and domestic water in the nine counties of the Liujiang River flows directly. The Liujiang River has also become the most important water area in Liuzhou, and it is also the final sewage body of the city’s industrial and agricultural wastewater and domestic sewage. The amount of wastewater discharged into the Liujiang River every year is as high as 350 million tons, of which industrial wastewater accounts for 83%. The fragility and industrial development of the ecological environment in the karst area, among other factors, have put the water and environmental safety of the Liujiang River under great pressure [12,29]. Previous studies found that the mass concentrations of heavy metal concentrations in the Liujiang River increased in 2013, 2015, and 2018 [24,26,27] (Table 1), indicating that the problem of heavy metal pollution in the rivers of the Liujiang River is relatively serious in general. Therefore, it is of great scientific significance to carry out studies on the pollution characteristics, spatial and temporal distribution patterns, and source analysis of water chemistry and heavy metals in the Liujiang River basin, providing background basic data for subsequent studies in the Liujiang River basin and providing a scientific basis for water quality safety in karst areas.

### 2.2. Water Sampling and Analytical Methods

A total of 111 water samples were collected in R1 and R2, which reasonably represent the water quality throughout the Liujiang River during 2019 (March, June, and November) (Figure 1). For each section (R1 and R2), day and night monitoring data of the hourly concentration of heavy metals were collected at frequencies of 00:00, 04:00, 08:00, 12:00, 16:00, and 20:00 or 02:00, 06:00, 10:00, 14:00, 18:00, and 22:00; 111 river samples were collected 5–7 times per month.

Water samples were collected in triplicate from surface water in cleaned polyethylene bottles from selected sampling sites in the study area. The pH, electrical conductivity (EC), dissolved oxygen (DO), and total dissolved solids (TDSs) were measured in situ using a portable analyzer (PONSEL, France). Free carbon dioxide levels were determined by alkaline titration. The HCO_3_^−^ concentrations were determined using an alkalinity kit (Merck) with an accuracy of 0.1 mmol/L. A 500 mL field-filtered water sample from a 0.45 μm acetate membrane was collected in a polyethylene bottle. Samples for Mg^2+^, Ca^2+^, Na^+^, K^+^, and heavy metal determinations were added to an appropriate amount of 1:1 HNO_3_ solution to make pH < 2. Samples for NO_3_^−^, SO_4_^2−^, and Cl^−^ determinations were left untreated. The water sample was stored in a 4 °C environment for subsequent detection.

NO_3_^−^, SO_4_^2−^, and Cl^−^ were measured by an ion chromatograph (861 Advanced Compact IC Metrohm, Swiss Confederation) with an accuracy of 0.01 mg/L. Mg^2+^, Ca^2+^, Na^+^, K^+^, and various heavy metals were determined by an ICP-OES spectrometer (IRIS Intrepid II XSP, Thermo Fisher Scientific, USA). Triplicate analyses of standard material were performed to check the precision within 5%. These data produced satisfactory results, with analytical errors within 1–10% for different elements.

### 2.3. Risk Assessment of Human Health

Heavy metals in drinking water enter the human body in two main ways, through ingestion and skin absorption [10,30,31]. Based on the recommendation of the US Environmental Protection Agency, the hazard quotient (HQ) and the hazard index (HI) are the key parameters that are used to measure the human health risk associated with consuming heavy-metal-contaminated water. The evaluation model referenced the literature [32] and a previous study [33].

Input parameters such as exposure time, exposure duration, ingestion rate, body weight, and surface area were considered as per the Chinese conditions and USEPA. To estimate the non-carcinogenic/chronic risk, chronic daily intake (CDI) and HQ can be calculated by Equations (1) to (2) [31,34].
(1)$$CDI_{\mathrm{ingestion}}=\frac{C_{W}\times IR\times EF\times ED}{B_{W}\times AT}$$
(2)$$CDI_{\mathrm{dermal}}=\frac{C_{W}\times SA\times Kp\times ET\times EF\times ED\times CF}{B_{W}\times AT}$$
where C_W_ is the individual metal concentration (µg/L), IR is the ingestion rate, EF is the exposure frequency, ED is the exposure duration, B_W_ is the body weight, AT is the average time (ED × 365 days/year), SA is the skin area, ET is the exposure time, CF is a unit transfer factor (Appendix A), and Kp (cm/h) is the dermal permeability coefficient of the metals in water (Appendix A) [31,35].

HQ denotes the noncarcinogenic risks. The HQ value is calculated by Equations (3) and (4).
(3)$${\;\mathrm{HQ}}_{\mathrm{ingestion/dermal}}=\frac{{\mathrm{CDI}}_{\mathrm{ingestion/dermal}}}{{\mathrm{RfD}}_{\mathrm{ingestion/dermal}}}$$

RfD_Ingestion_ and RfD_Dermal_ are the chronic reference doses of ingestion and dermal (µg/kg/day), respectively (Appendix A). If the HQ exceeds 1, there might be concern for noncarcinogenic effects.

To evaluate the total potential noncarcinogenic risks posed by more than one pathway, the HI was calculated to account for the total noncarcinogenic risk of all individual heavy metals combined in each exposure pathway (Equation (5)). The total hazard index (THI) was calculated by summing the HIs in each exposure pathway (Equations (6) and (7)).
(4)$$THI=HI_{\mathrm{ingestion}}+HI_{\mathrm{dermal}}$$
(5)$$HI_{\mathrm{ingestion/dermal}}=\sum_{I=1}^{n}HQ_{\mathrm{ingestion/dermal}}$$
(6)$${\mathrm{THI}}_{\left(\mathrm{adult}\right)}={\mathrm{HI}}_{\mathrm{ingestion}\left(\mathrm{adult}\right)}+{\mathrm{HI}}_{\mathrm{dermal}\left(\mathrm{adult}\right)}$$
(7)$${\;\mathrm{THI}}_{\left(\mathrm{children}\right)}=HI_{\mathrm{ingestion}\left(\mathrm{children}\right)}+HI_{\mathrm{dermal}\left(\mathrm{children}\right)}$$

HI and THI values > 1 indicated a potential for an adverse effect on human health or the necessity for further study [31].

The total cancer risks (CR) caused by different pathways (Equation (8)) were then evaluated [33,36]. The total cancer risk (TCR) was obtained by the sum of CRs calculated for two pathways (Equation (9)).
(8)CR=∑CDI×SF
(9)$$TCR=CR_{\mathrm{ingestion}}+CR_{\mathrm{dermal}}$$

Cd, As, Ni, Pb, and Cr are chemical carcinogens, and Cu, Mn, Co, Al, Hg, and Zn are chemical noncarcinogens. SF is the reference dose (mg/(kg·d)^−1^) and represents the slope factor of heavy metals (Appendix A). The RfD and SF values were mainly collected from the RAIS and IRIS databases [37,38].

## 3. Results

### 3.1. Basic Physical and Chemical Properties of Water

The pH value of the water body in the Liuzhou section varied from 6.70 to 7.35, with an average value of 7.09. The overall pH value was weakly alkaline, which is consistent with the characteristics of rivers in karst areas [39]. In terms of seasonal changes, the pH in November and March were relatively close, with the highest in November and the lowest in June (Table 2). The average pH value of the river bottom was 3.79 [40], which is quite different from the pH value in the water body. Some studies have shown that the content of CO_2_ in water is one of the factors that determines the pH value of the water body [41]. It can be seen that pH and free CO_2_ had a negative linear relationship (Figure 2), so rainfall may bring CO_2_ in the atmosphere into the water body, and CO_2_ mainly entered the water body through the carbonic acid balance equation: CO_2_ + H_2_O → H_2_CO_3_, H_2_CO_3_ → H^+^ + HCO_3_^−^ produces H^+^ [42], resulting in lower pH in seasons with high rainfall.

The EC and TDSs reflected the content of soluble ions and soluble substances in water, respectively. The TDS of river water samples ranged from 114.17 to 164.05 mg/L, with an average value of 140.27 mg/L, which is lower than the average TDS value of the Xijiang River, which was 174 mg/L [43], i.e., much higher than the world river average of 65 mg/L [44]. A higher EC reflects higher water pollution [45], and the conductivity value of the Liujiang water body was lower than the maximum allowable conductivity value (250 μs/cm) in drinking water specified by the WHO.

Temperature (T) has a key influence and regulates the life activities of animals and plants in water. The time of regulation was ranked as June > November > March, which relates to the subtropical monsoon climate in Liuzhou.

Dissolved oxygen (DO) is an important indicator to measure the self-purification ability of water bodies. Under normal circumstances, the dissolved oxygen in the water body was 5–10 mg/L. The average mass concentrations of dissolved oxygen in R1 and R2 were 6.84 mg/L and 7.96 mg/L, respectively, indicating that the self-purification capacity of the water body is high.

Turbidity is a physical and chemical parameter with the largest variability. The variability of turbidity in June exceeded 200 (Table 2), which is a strong variable. This may be caused by the agitation of the river sediment caused by the heavy rainfall in June.

### 3.2. Major Ions and Heavy Metals in River Water

Major ions and heavy metals of Liujiang River water along temporal scales are described in Figure 3, Figure 4 and Figure 5. For all the monthly samples, from determining the main water chemical ion content using Piper three-line graphs [46], the results show that Ca^2+^ was the main ion in the cation (78% of the total cations), and the anions were dominated by HCO_3_^−^ (86% of the total anions) (Figure 3). The water chemistry type of the Liujiang River was the Ca^2+^-HCO_3_^−^ type [47], mainly controlled by carbonate weathering. The concentrations of major ions and heavy metals displayed great seasonality. The peak values of HCO_3_^−^, Ca^2+^, Mg^2+^, Cl^−^, NO_3_^−^, Cd, Cr, Mn, Ni, Pb, and Hg appeared in March; pH, K^+^, Na^+^, SO_4_^2−^, As, Zn, and Cu appeared in November; and Al appeared in June (Figure 4 and Figure 5).

### 3.3. Pollution Source Analysis

#### 3.3.1. Analysis Procedures

In this paper, correlation and redundancy analyses were conducted between anions and cations (HCO^3−^, K^+^, Na^+^, Ca^2+^, Mg^2+^, Cl^−^, SO_4_^2−^, and NO_3_^−^) and heavy metals (As, Cd, Zn, Cr, Cu, Mn, Ni, Co, Al, Pb, and Hg) and environmental factors (pH, TDS, turbidity, DO, and pH) in water bodies. The correlation coefficients ranged from −1 to 1. The closer the correlation coefficient was to 1 or −1, the stronger the correlation; the closer the correlation coefficient was to 0, the weaker the correlation. The correlation coefficient was greater than 0 for positive correlations, less than 0 for negative correlations, and equal to 0 for zero correlations. RDA looks for the relationship between changes in anions and cations and heavy metals and environmental factors by ranking. Anions and cations were used as response variables for the ranking analysis and hydrodynamic conditions as explanatory variables [48]. Detrending analysis (DCA) of heavy metals showed that the response data were combined, with a maximum value of 0.3 < 3 for the gradient length of each ordination axis, so a redundancy analysis using a linear approach was chosen. The further the projection points on the RDA ordination plot fall in the same direction as the arrow indicates, the higher the correlation, with projection points near the origin of the coordinates (zero points), indicating that the correlation was close to zero. If the projection points lie in opposite directions, the predicted correlation is negative.

#### 3.3.2. Pearson Correlation and RDA Analysis

Pearson correlation-based heatmaps graphically demonstrated that the distribution of heavy metals and ions was strongly affected by the basic physical and chemical properties of water (Figure 6A,B). The main water physicochemical parameters affecting anions and heavy metals in the two seasons were pH, EC, and TDS. During the dry season, the influence of water body acceptance properties was more significant. RDA plots on the dry season and wet season water quality parameters were performed to explore the key factors affecting water quality changes in different seasons (Figure 6C,D). According to the RDA plots, the first two axes explained 83.64% and 89.56% of the variability in the wet season and dry season water quality, respectively. The major environmental variables driving wet season water quality were TDS (F = 25.6, *p* = 0.002) and pH (F = 6.0, *p* = 0.002), and the dry season was significantly correlated with TDS (F = 42.5, *p* = 0.002) and pH (F = 4.0, *p* = 0.006).

### 3.4. Risk Assessment of Heavy Metals in River

The results for noncarcinogenic risks are shown in Appendix A and Figure 7. The health risks of noncarcinogenic heavy metals in the dry season and wet season of the Liujiang River are presented in Appendix A and Appendix A, respectively. In the dry season, the HI_ingestion_ and HI_dermal_ of all elements in adults and children at R1 were below 1. However, the HI_ingestion_ of Al at R2 was exceeded 1 for children, and the HI of Al for adults was near 1 at R2 (Appendix A). The THIs in adults and children in the river water of R1 and R2 were 5.5 × 10^−1^, 7.7 × 10^−1^, 1.13 × 10^0^, and 1.59 × 10^0^, respectively (Figure 7). During the wet season, the HI_ingestion_ of Al at the sampling station (R1 and R2) was above 1 for adults and children, followed by As, Cr, and Cu, while HIdermal of elements for both adults and children was below 1 (Appendix A). The THIs of adults and children in R1 were 7.49 × 10^0^ and 1.16 × 10^0^, respectively, while in R2, they were 6.77 × 10^0^ and 9.45 × 10^0^, respectively (Figure 7). CRs were estimated for the heavy metals (i.e., Cd, As, Cr, Ni, and Pb) in Liujiang river (Appendix A, Figure 8). Similar to the noncarcinogenic risk, the ingestion of CR was much higher than the dermal results (Appendix A). In addition, each metal had a lower TCR in adults than in children (Figure 8).

## 4. Discussion

### 4.1. Characteristics of Changes in the Concentration of Major Ions Heavy Metals

Major ions and heavy metals of Liujiang River water along temporal scales are described in Figure 3, Figure 4 and Figure 5. Regarding the two sampling months in the dry season, the diluting effect in the dry season did not result in the consequent reduction in the concentrations of Mg^2+^, Cl^−^, Cd, Cr, Mn, Ni, and Pb. During the continuous rainfall in March, the concentrations of Mg^2+^, Cl^−^, Cd, Cr, Mn, Ni, and Pb remained stable or increased, while NO_3_^−^, HCO_3_^−^, and Ca^2+^ showed fluctuations with rainfall (Figure 4 and Figure 5). Studies have shown that Liuzhou can sow crops such as rice, corn, and vegetables on a large scale after mid-March [49]. During the sowing and growing seasons of crops, it is speculated that a large amount of fertilization and industrial activities combined with abundant precipitation led to an increase in the concentration of major ions and heavy metals in the river water, and agricultural activities significantly increased the NO_3_^−^ concentration. The high coefficient of variation of calcium ions and bicarbonate may be related to the dissolution of carbonate rocks in the study area. In November, the low temperature and precipitation were the most plausible explications for pH, K^+^, Na^+^, SO_4_^2−^, As, and Cu concentrations remaining steady. Furthermore, the mass concentration of Zn had an abnormally high value, increasing anthropogenic activities such as agriculture and mineral processes in the summer elevated the total concentrations of Zn [50], and the highest concentrations of ions and heavy metals in March and November were due to intense anthropogenic activities (agriculture and industrial activities).

The highest total concentrations in June were due to the high precipitation in summer (Figure 4 and Figure 5). The diluting effect in the wet season due to high precipitation resulted in Cd, Zn, Co, and Hg elements not being detected and more variables having their lowest concentrations (Figure 5). Regarding the sampling month in the wet season, large storms in the Liujiang River could initially increase the concentrations of variables, as the mixing of large volumes of noncontaminated runoff water, most individual ions, and metal content tended to decrease (Figure 4 and Figure 5). The mass concentration of the A1 element rose and fell sharply in June, and the coefficient of variation was relatively high. The mass concentrations of Al corresponded to the magnitude of the instantaneous rainfall and respond quickly to rainfall. The high concentrations of the A1 element were predicted to be caused by the leaching and scouring effect of the instantaneous rainfall being released in a short time. Al showed significant differences between the wet (April–September) and dry (October–March) seasons. In general, larger quantities of water in the wet season would lower the concentrations. Among the 11 heavy metals, only Al exhibited a higher concentration in the wet season, indicating that the input of Al was a combined result of natural and anthropogenic sources. Al had higher concentrations in R2 than in R1 (Figure 5). The sampling point R2 was located in the main industrial zone of the study area, and there were multiple industrial parks. There may be some input points of this area, such as the metallurgy of Al, mining, and other industrial sewage discharge. Therefore, further monitoring of Al should be continued and strengthened.

Various metals showed higher compositions in the dry season, meaning that the risk of dry season may be greater. In the industrial area (R2) especially, Mn, Ni, Al, Cr, Hg, and As concentrations were found to be higher in R2 than in R1, which resulted from seasonal anthropogenic activities [51].

Table 3 provides a comparison of the metals in the river of the Liujiang River with the guideline values and other studies. The average concentrations of heavy metals in the dry and wet seasons were in the order of Al > Zn > As > Cr > Ni > Cu > Mn > Pb > Hg > Cd > Co, and Al > Cu > Mn > As > Cr > Pb = Hg. Cu, Mn, and Al exhibited significantly higher concentrations in the wet season than in the dry season, and Cd and Ni were higher in the dry season than in the wet season. These findings were similar to the results reported in the Hanjiang River. This phenomenon is related to the precipitation and seasonal anthropogenic activities on the Liujiang and Hanjiang Rivers [52].

The data compared with other rivers of China reveal that concentrations of Cd were higher than those of the Jiulong River, which was primarily due to dominant physical weathering in the Jiulong River. Liujiang and Jiulongjiang are also located in economic cities, but the sources of Cd are different. The source of Cd in the Jiulong River water body is closely related to soil parent material, geochemistry, and agricultural production activities [53]. The Liujiang River is mainly affected by industrial activities, and the difference in Cd content is caused by different levels and methods of human activities. Lower than that of the large watersheds, such as the Pearl River and Yangtze River (where large populations and industries are concentrated), urbanization and industrialization have caused high levels of metal pollutants in the Pearl River and the Yangtze River, where large populations and industries are concentrated in the study area and create large amounts of residual anthropogenic metals [54].

Comparing the heavy metal content of the Liujiang River with that of the world’s rivers, it was found that the heavy metal concentrations in the water of our study area were much lower than those of the reference rivers in Africa [55] and South America [56]. The lower intensity of industrial and domestic activities compared to South America over China’s large river drainage basins caused low concentrations of dissolved heavy metals from large rivers in China. The heavy metal data in Table 2 are comparable with the results from small and less disturbed world rivers, such as the Yamuna and Danube [57,58], mainly due to heavy metal pollution of the river caused by artificial activities in the surrounding areas.

The mean concentrations of all heavy metals in the Liujiang River were much lower than the drinking water guideline values set by the WHO (2011) [59] and the USEPA (2006) (Table 3) [60]. Cd in the dry season was higher than the criterion continuous concentration (CCC) values of the USEPA water quality criteria (Table 2). The dry season mercury levels are above the threshold of the Chinese standard GB3838-2002 for tertiary water quality. The relatively high mass concentration of Hg in the drinking water source of the Liujiang River Basin may be related to the widespread use of pesticides in the 1980s [26], mainly due to the high regional background value caused using pesticides. In addition, the average metal concentrations in Liujiang water were above the world average background values and below other water quality standards or rivers, so our results were comparable to those of uncontaminated rivers (Table 2).

**Table 3 ijerph-19-15435-t003:** Analysis of variance for heavy metal concentrations in the dry and wet seasons in the Liujiang River and comparison with other studies and guidelines (unit in μg/L).

Statistics	As	Cd	Zn	Cr	Cu	Mn	Ni	Co	Al	Pb	Hg	Data Sources
Liujiang river, dry season	1.38	0.26	6.75	1.00	0.64	0.61	0.95	0.11	7.73	0.35	0.26	This study
Liujiang river, wet season	0.67	N.D.	N.D.	0.66	0.76	0.73	0.83	N.D.	83.76	0.10	0.10	This study
the upper Han River, dry season	10.74	3.21	N.D.	5.89	7.76	42.2	1.15	N.D.	110.86	7.37	N.D.	[52]
the upper Han River, wet season	17.58	1.38	N.D.	10.32	18.97	18.79	2.26	N.D.	262.42	11.02	N.D.	[52]
Jiulong River	12.39	0.08	154.89	5.41	17.85	N.D.	3.99	N.D.	N.D.	4.47	N.D.	[53]
Pearl River	N.D.	2.6	22.2	3.5	11.1	116	12.8	5.5	N.D.	18.5	N.D.	[61]
Yangtze River	13.2	4.7	N.D.	20.9	10.7	5.4	13.4	N.D.	N.D.	55.1	N.D.	[54]
Nile River, Africa	N.D.	7.67	351.67	25.50	33.17	61.85	N.D.	N.D.	N.D.	27.16	0.98	[55]
Danube River, Europe	N.D.	0.07	23.03	0.83	3.22	N.D.	1.80	N.D.	N.D.	0.84	N.D.	[57]
Matanza-Riachuelo River, South America	N.D.	17.33	98.67	33.33	N.D.	123	52.67	N.D.	N.D.	118.33	N.D.	[56]
River Yamuna, India	N.D.	0.05	1.50	0.15	2.15	N.D.	0.37	N.D.	N.D.	0.11	N.D.	[58]
Background word average		0.08		1.00		0.3	0.2					[62]
The National Surface Water Environmental Quality Standards (GB3838-2002) in China	50	5	1000	50	1000					50	0.1	[63]
Freshwater quality criteria for protection of aquatic life												
CMC, acute	340	2.0		16	13		470		750	65		[60]
CCC, chronic	150	0.25		11	9		52		87	2.5		[60]
Water quality criteria for drinking water												
WHO	10	3	3000	50	2000	400	20		200	10	1	[59]
China	10	5	1000	5	1000	100	20		200	10	1	[64]
US EPA, MCLG	0	5		100	1300					0	2	[60]
US EPA, MCL	10	5		100	1300					15	2	[60]

The dry season includes March and November, and the rainy season includes June; CMC, criterion maximum concentration; CCC, criterion continuous concentration; MCLG, maximum contaminant level goal; MCL, maximum contaminant level. N.D. indicates no detection.

### 4.2. Correlation and RDA Source Analysis of Major Ions and Heavy Metals in River

The results of the major ions and heavy metals correlation analysis and RDA source analysis are shown in Figure 6. Water environmental factors influenced the contents of heavy metals, anions, and cations. In the wet season, As and Cu were positively correlated with turbidity, indicating that As and Cu mainly exist in the form of particles, and the presence of suspended particles can significantly increase the concentration of As and Cu in water. As they displayed a positive correlation with Cu, we ascribed them to anthropogenic inputs such as production and domestic wastewater. HCO_3_^−^, Ca^2+^, Mg^2+^, Cl^−^, and SO_4_^2−^ were significantly affected by TDS, which was mainly related to the weathering of the basin rocks. Cr, Ni, and K^+^ were significantly affected by EC, indicating that Cr and Ni had ionic forms in water, and K^+^ had no significant positive correlation with other ions. Cr, Ni, and K^+^ might be related to exogenous industrial and mining discharge. There was a positive correlation between Na^+^ and pH, indicating that the alkaline environment is favorable for the existence of Na^+^. Mn, Al, and NO_3_^−^ were significantly affected by DO, indicating that the presence of DO in water promotes the dissolution of Mn, Al, and NO_3_^−^, which is influenced by nonpoint sources such as agricultural runoff.

During the dry season, HCO_3_^−^, Ca^2+^, Mg^2+^, K^+^, Ni, Co, and Cr were significantly affected by TDS and EC. There was a positive correlation between HCO_3_^−^, Ca^2+^, Mg^2+^, and K^+^, which could be due to the leaching of bed rocks, and because carbonate rocks are the dominant lithology in the upper and middle reaches of the Liujiang Basin of China [23,65]. Ni displayed a positive correlation with Co, Cr, and Cu, and the accumulation of Co, Cr, and Cu in Liujiang River water was inevitably affected by the mining operations taking place upstream of the Liujiang River and Liuzhou Urban area industrial operations related to metal smelting and chemicals, food, and paper industries [33,40]. The correlation between Al, Na^+^, and Cl^−^ was more significant; the correlation between Na^+^ and K^+^ was not significant; and K^+^ was mainly related to geological activities, which means that Na^+^ is mainly affected by human activities. The Cl^−^ mass concentration in rainfall (approximately 0.47 mg/L on average) was much smaller than the Cl^−^ mass concentration in rivers (approximately 3.54 mg/L on average), so Cl^−^ is also very likely to come from the input of human activities, which may be related to the discharge of domestic wastewater (washing clothes), and Al is also closely related to human production and life. SO_4_^2−^ and NO_3_^−^ were significantly affected by pH, and Zn, Cu, and Hg were significantly positively correlated with DO, which may be the contribution of precipitation to the input of exogenous Zn, Cu, Hg, and DO, as well as the runoff, infiltration, and transport brought by precipitation. The main route for the release of Zn, Cu, and Hg was from soil to water. Mn was positively correlated with turbidity, and turbidity was mainly related to the disturbance of the river. The disturbance of the river may cause the release of Mn in the sediments and increase the dissolved Mn content in the water body.

### 4.3. Human Health Risk Assessment of Heavy Metals

#### 4.3.1. Noncarcinogenic Risk Assessment

Various routes and contributions of different metals to HI are shown in Appendix A. Ingestion was an important route of exposure to heavy metals in the sample station, especially in the wet season (Appendix A). In both seasons, we conclude that the four factors contributing most to chronic risk in adults and children were Al, followed by As, Cr, and Cu, while the smallest contributions were Zn, Ni, and Mn (Appendix A). The health risks of noncarcinogenic heavy metals in the dry season and wet season of the Liujiang River are presented in Appendix A and Figure 7, respectively. In the dry season, the HIingestion of Al at R2 exceeded 1 for children. During the wet season, the HIingestion of Al at the sampling station (R1 and R2) was above 1 for adults and children, implying that Al may cause adverse health effects and potential nocarcinogenic concern. Figure 7 shows that the obtained THIChildren and THIAdults at sampling sites R2 were >1, indicating that R2 is unsafe for adult and child human health [66]. The THIChildren values of sampling sites R1 were >1, indicating that R1 is unsafe for human health in children. Thus, there is a need to improve the safe management of drinking water for children in the Liujiang River. The average total health risk of noncarcinogenic heavy metals in the wet season was slightly higher than that in the dry season. This is because of the lower water quantity in the dry season than in the wet season, and some heavy metals could be washed through the wet season water and finally dissolved in the rivers. From the contribution of different metals to HI at the sample station, in the wet season, Al contributed 98.72% to the noncarcinogenic risk in R1. Likewise, in R2, Al contributed 98.66%. In the dry season, the noncarcinogenic contributions in R1 and R2 through Al reached 65.64% and 81.79%, respectively (Appendix A). Al was not only the heavy metal with the highest noncarcinogenic risk, but its concentration was positively correlated with rainfall (Figure 4). Thus, the higher noncarcinogenic risk during the rainy season could be related to rainfall.

#### 4.3.2. Carcinogenic Risk Assessment

The CR and TCR values between 10^−4^ and 10^−6^ indicate that carcinogenic risk is considered an acceptable range. CR and TCR values <10^−6^ indicate that carcinogenic risk is considered negligible, while values >10^−4^ indicate that carcinogenic risk is considered unacceptable [31,32,67,68]. As exhibited the highest health risk among the five heavy metals in the Liujiang River in the dry season, followed by Ni, Cd, Cr, and Pb. For children, the health risk of As was higher than the maximum acceptable risk level of 1 × 10^−4^. For the two age groups, Ni, Cd, and Cr were at acceptable risk levels, and Pb was at no risk (Figure 8). As, Cr, Cd, and Ni were mainly distributed in the suburban section of Liuzhou such as Liujiang downstream (R2), and source analysis shows that Cr, As, and Ni may come from industrial waste water in the dry season. The carcinogenic risk of R2 was higher than that of R1 mainly because R2 was located in an industrial concentration area and was greatly affected by industrial activities.

The TCR values of this study show that dry season water was more risky than wet season water, and the maximum carcinogenic risks of Cr, As, Cd, Ni, and Pb were in the dry season because of high metal concentrations due to low water concentrations. For example, Cd was not detected during the wet season through the erosion and dilution effect of rainfall, resulting in a lower risk of carcinogenesis in the rainy season.

Health risk evaluation of the Liujiang River revealed that the water at the sampling station at the downstream Luowei section was unsafe for drinking or living. This risk varies seasonally, with carcinogenic risk in the dry season and noncarcinogenic risk in the wet season. Through this study, it was observed that the mere monitoring of elemental metal concentrations in one season (dry season or wet season) will not provide a clear idea about serious health hazard conditions. The risk of health hazards in this study was only shown through the drinking water and skin contact route into the human body, but in fact there are many other ways for toxins to enter the body, such as through food, air, etc., so the actual risk of health hazards of various chemical substances is much greater than this. Therefore, research on the health risk assessment of heavy metals in different water level operation periods of the Liujiang River needs further exploration

## 5. Conclusions

Concentrations of major ions (HCO_3_^−^, Ca^2+^, Mg^2+^, Cl^−^, NO_3_^−^, K^+^, Na^+^, and SO_4_^2−^) and dissolved heavy metals (Al, As, Cd, Co, Cr, Cu, Mn, Ni, Pb, Hg, and Zn) in the surface water of the Liujiang River demonstrated great seasonality. The major ion and heavy metal concentrations in the wet season water were slightly lower than those in the dry season water in the Liujiang River. Al exhibited a higher concentration in the wet season because it responds quickly to rainfall. Heavy metals showed distinct regional variations and were mainly distributed in downstream urban sections of the Liuzhou area. The concentrations of Hg in water samples were above the limits recommended by China’s surface water environmental quality standard (GB 3838-2002). Source analysis of the Liujiang River Basin shows that there are geological and anthropogenic sources of heavy metals in the river, with anthropogenic industrial sources having the greatest impact. Human health risk assessment results indicate that the average total health risk was higher for children than for adults. The average noncarcinogenic health risk of heavy metals in the wet season was higher than that in the dry season. The HI values of Al in both seasons were higher than the noncarcinogenic limiting value (HI ≤ 1). Al was the most important pollutant causing noncarcinogenic effects. The carcinogenic health risk of heavy metals in the dry season was slightly higher than that in the wet season. As was the most important pollutant causing carcinogenic concerns, particularly in R2. Al and As were the main causes of health risk in the aquatic environment of the Liujiang River area and should be prioritized as the main objects of aquatic environment risk management in the Liuzhou area. The Liujiang downstrem (R2) should be particularly ranked as the key prevention and control area. The exposure concentrations and health effects of As and Al for both seasons in the Liujiang River should be addressed further.

## Figures and Tables

**Figure 1 ijerph-19-15435-f001:**
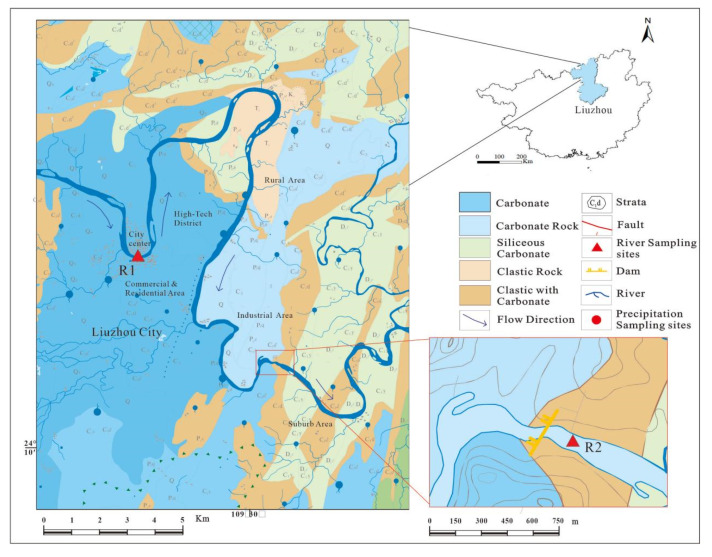
Schematic map of the Liujiang Basin and the sampling sites.

**Figure 2 ijerph-19-15435-f002:**
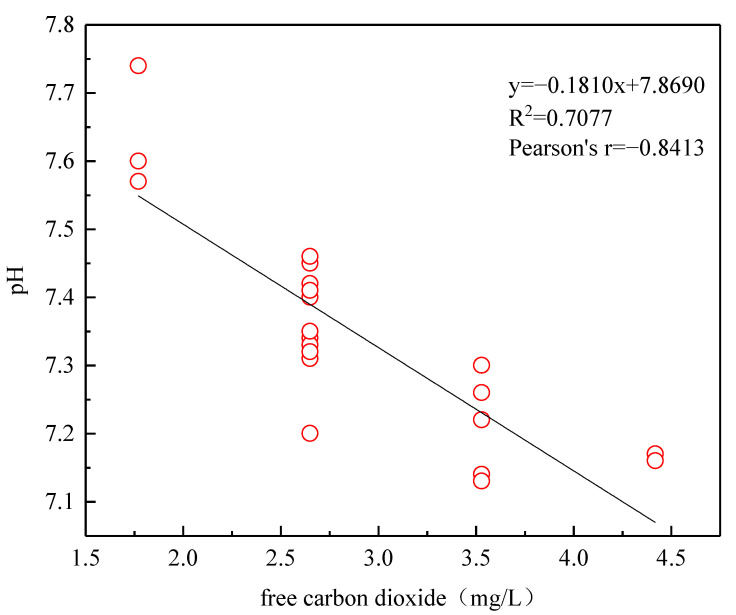
Linear fitting of free carbon dioxide and pH.

**Figure 3 ijerph-19-15435-f003:**
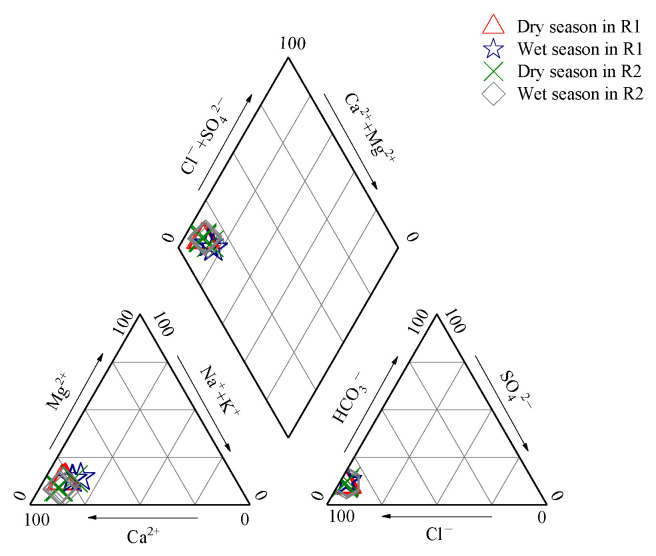
Piper chart of river water in the study area (ion equivalent concentration).

**Figure 4 ijerph-19-15435-f004:**
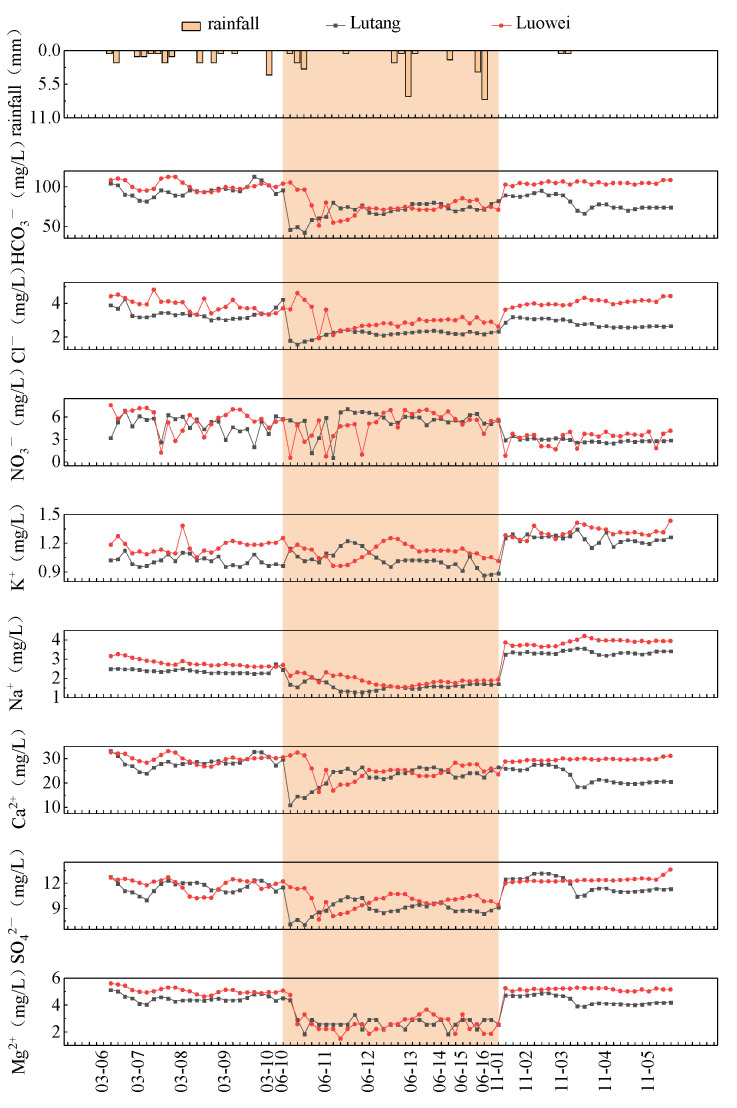
Average concentration variations of ions at different water level operation periods.

**Figure 5 ijerph-19-15435-f005:**
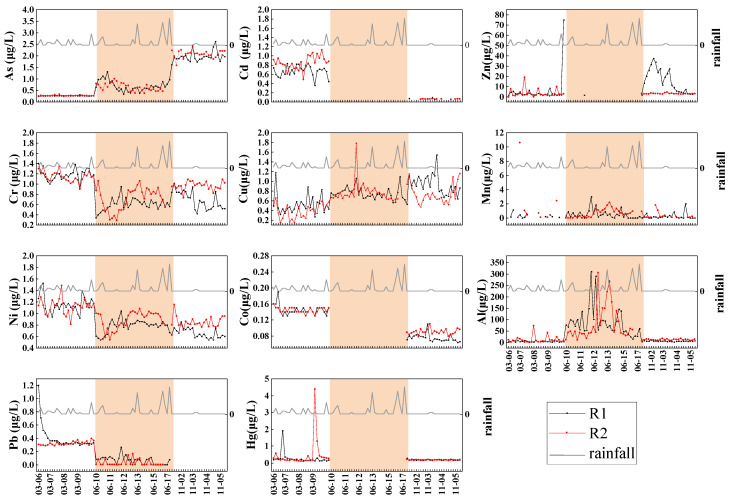
Average concentration variations of heavy metals at different water level operation periods.

**Figure 6 ijerph-19-15435-f006:**
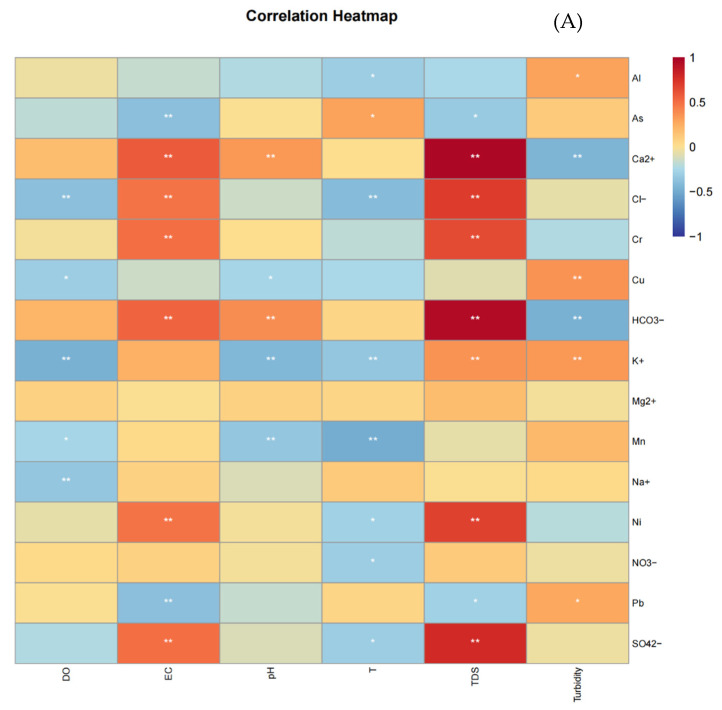
Influencing **factors** for selected parameters analyzed in the water samples collected from the Liujiang River in the wet season (**A**,**C**) and dry season (**B**,**D**) in China.

**Figure 7 ijerph-19-15435-f007:**
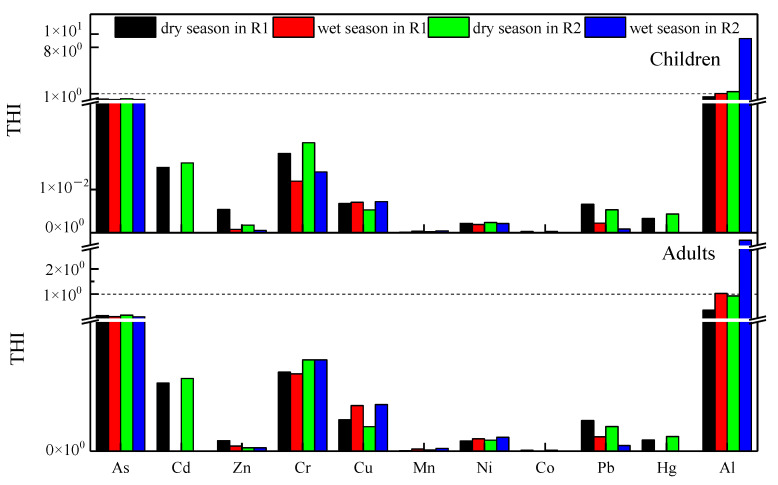
Total hazard index (THI) evaluation in the sampling sites for selective aged groups.

**Figure 8 ijerph-19-15435-f008:**
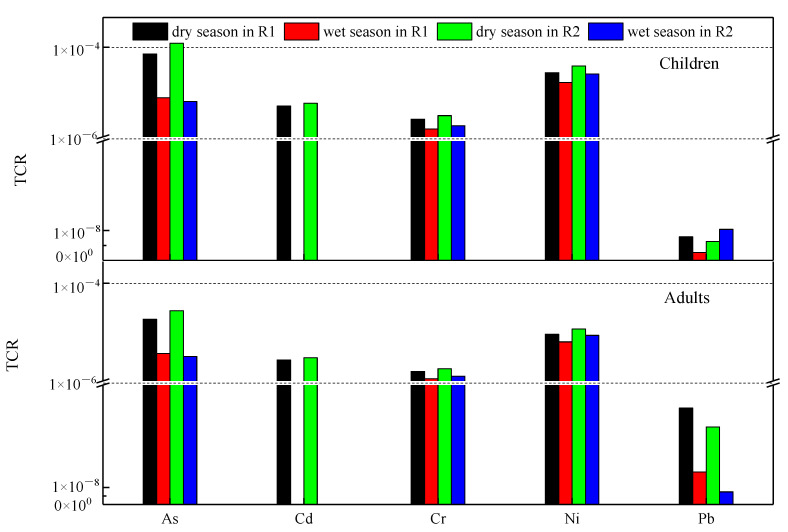
Total carcinogenic risk (TCR) evaluation in the sampling sites for selective aged groups.

**Table 1 ijerph-19-15435-t001:** Liuzhou annual average concentration of heavy metal pollutant statistics (μg/L).

Year	Cd	As	Cr	Hg	Pb	Cu	Ni	Zn	Mn
2013	1.269	1.25 × 10^−4^	0.672	0	14.167	3.969	-	-	-
2015	0.02	1.15	1.16	-	0.29	1.02	0.7	1.87	-
2018	1.28	1.28	2.88	0.02	2.45	16.83	-	6	118.17

**Table 2 ijerph-19-15435-t002:** Basic physical and chemical parameters of the Liujiang water body.

Month	Sampling Point		pH	TDS	T (℃)	DO (mg/L)	EC (μs/cm)	Turbidity (mV)
March	R1	Average	7.01	151.06	13.33	10.55	189.05	31.75
		Variation	0.09	9.56	0.34	0.28	10.79	9.63
	R2	Average	7.29	161.53	12.31	10.31	204.6	44.58
		Variation	0.09	9.56	0.34	0.28	10.79	9.63
June	R1	Average	6.84	114.17	24.64	8.26	148.86	294.63
		Variation	0.14	14.51	0.29	0.44	20.09	208.6
	R2	Average	6.69	122.33	24.09	7.53	165.21	343.32
		Variation	0.52	17	0.54	0.38	16.66	232.89
November	R1	Average	7.35	128.51	14.65	10.84	176.01	1.48
		Variation	0.11	13.51	0.57	0.29	31.8	0.27
	R2	Average	7.34	164.05	13.65	8.5	199.81	1.38
		Variation	0.15	3.13	0.57	0.23	11.95	0.08

## Data Availability

Not applicable.

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
