# Peer review of "Spatial Distribution, Source Analysis and Health Risk Study of Heavy Metals in the Liujiang River Basin in Different Seasons"

_ijerph, 2022, doi:10.3390/ijerph192315435_

Round 1

Reviewer 1 Report

Thank you for giving me the opportunity to review this manuscript. Overall, the paper is well written and structured, but I have few minor comments that I wish the authors could address them.

1. I believe the results and discussion should be separated and should not be combined together. The current combination of the results-discussion has disabled some further elaborations on temporality/plausibility/consistency of the findings with other literature.

2. Authors should consider to build a suitable cartography to highlight the concentration of the pollutants found in the results part. This will enable proper "visualization" of the spatial distribution.

3. It is unclear multivariate analyses included what tests. Probably the methodology reporting could be improved a little to include a subsection on "analysis procedures." 

4. Also, in the methodology, please include a table if appropriate on the normal concentrations of pollutants/chemicals anticipated in those areas. This would justify on why this study should be conducted further.

Reviewer 2 Report

This paper described spatial and temporal analysis of heavy metal concentration in Liujang River Basin, China and its health risk. Overall, the manuscript is OK, however, there are a few issues/ comments need to be addressed before it can be accepted:

1. Title - the title include source identification but it the results presented in abstract, discussion and conclusion did not addressed this. In addition, the objectives listed in introduction did not cover source identification. Please revise.

2. Method - the method was unclear. In the results, Pearson correlation was presented, however, no description on Pearson in method was given. Plus, from the heat map (Figure 6), there were clustering arrow at the left hand side of the figure, however, it was not discussed. Did the authors embedded clustering method as well? The formula for health risk was incomplete. The was no formula for THIadult and THIchildren given in the method, however, it was reported in the results. The details comment on this section can be referred in the attached document. 
